



# Some comments on experimental results of three lift controllers for a wind turbine blade section using an active flow control

Loïc Michel[1], Caroline Braud[2], Jean-Pierre Barbot[1,3], Franck Plestan[1], Dimitri Peaucelle[4], and Xavier Boucher[5]

[1]Nantes Université, Ecole Centrale Nantes, CNRS, LS2N, UMR 6004, 44000 Nantes, Nantes, France
[2]Nantes Université, Ecole Centrale Nantes, CNRS, LHEEA, UMR 6598, 44000 Nantes, Nantes, France
[3]ENSEA, Quartz Laboratory, EA 7393, 95014 Cergy-Pontoise, Cergy-Pontoise, France
[4]LAAS-CNRS, Université de Toulouse, CNRS, Toulouse, France
[5]Université du Québec à Trois-Rivières-LSSI, Trois-Rivières, Québec, Canada

**Correspondence:** Caroline Braud (caroline.braud@ec-nantes.fr)

**Abstract.** Controlling wind turbines is generally performed globally (rotor yaw or blade pitch control) to optimize the energy extraction and minimize rotor's loads for rotor's lifetime extension. This means that no information from the blade aerodynamics is up to now taken into account in the control loop while it is well understood that wind inflow interaction with blade aerodynamics can lead to power loss, load fluctuations and noise generation. This work deals with the development of control algorithms applied at the level of the blade section, considering only local aerodynamic sensing and actuators. The objective is to extract the maximum power from the wind energy by maintaining the aerodynamic lift at its highest value, while limiting load fluctuations using different solutions of control which take into account disturbances from different turbulent inflows. Some control strategies are investigated thanks to an experimental bench in the aerodynamic wind tunnel of LHEEA's laboratory in order to compare the tracking performances with respect to different operating scenarios.

## 1 Introduction

Controlling wind turbines is generally performed globally (rotor yaw or blade pitch control) to optimize the energy extraction or minimize rotor's loads for rotor's lifetime extension. This means that no information from the blade aerodynamics is up to now taken into account in the control loop while it is well understood that wind inflow interaction with blade aerodynamics can lead to power loss, load fluctuations and noise generation, see e.g. (Wagner et al. (1996); Rezaeiha et al. (2017)). Wind turbines are exposed to inflow turbulences of different scales due to the atmosphere in which they operate (see e.g. Schepers et al. (2021)) and also to rotor misalignment with the inflow or wakes of neighboring turbines. This is even more significant for offshore wind turbines whose rotor diameter are significantly larger, with local shear inflow over the rotor sweep area and even along the blades. To alleviate loads, the pitch control (Bossanyi (2000)) can be complemented by local and sometimes faster aerodynamic controllers. Local actuator types (such as vortex generators, flaps, slats, micro-jets / plasma) and sensor types (e.g. e-penons) have been developed for that purpose. Few contributions of control algorithms sufficiently robust to operate on the wind turbine blade aerodynamics have been proposed so far. Particularly, a significant number of controllers were investigated





for NACA profiles with objectives towards aeronautic applications, see e.g. (Becker et al. (2007)). Recently, (Bartholomay et al. (2021)) developed a feed-forward controller to alleviate the lift fluctuations under a well controlled sinusoidal inflow. (Jaunet and Braud (2018)) also developed a simple proportional controller to reduce lift fluctuations in a constant shear inflow
simulated by oscillations of a 2D blade section using fast control technologies like micro-jets.

This experimental work extends the active flow control methodology using micro-jets (Jaunet and Braud (2018)) towards controllers that are more suited against local perturbations at the blade scale. This work deals with the development of control algorithms applied at the level of the blade section, considering only *local* aerodynamic sensors and actuators (e.g. wall pressure sensors versus micro-jets and lift force versus blade pitch control). The objective is to extract the maximum power from the
wind energy and consequently to maintain the aerodynamic lift at its highest value, using different solutions of control and taking into account disturbances caused by turbulent perturbations (or inflows).

From previous works (Jaunet and Braud (2018); Peaucelle et al. (2019); Michel et al. (2022, 2024)), some control strategies are currently investigated to control the lift using micro-jets installed at the trailing edge of a 2D blade section that is mounted in a wind tunnel, thanks to an experimental bench in the aerodynamic wind tunnel of LHEEA's laboratory. The difficulty
in implementing this control lies in the very uncertain modeling of the lift, which naturally leads to the consideration of "model-free" type control laws. In this work, three control laws are investigated: a robust PID controller (Conord and Peaucelle (2021)), a *model-free control* (Fliess and Join (2013, 2021)), an adaptive control law based on the super-twisting algorithm (Shtessel et al. (2023)). The robust PID controller has been implemented considering a rough modeling of the lift response under particular parametric operating conditions of the test-bed.

The purpose of this study is to evaluate the performances of some selected control strategies under different operating conditions in order to characterize operating domain of each control law regarding some criteria like the nominal lift responses, the rejection of high frequency fluctuations, and the robustness with respect to modifications on the dynamics due to changes of the air flow characteristics.

The experimental bench used to test the efficiency of the control to alleviate aerodynamic load fluctuations includes a
perturbation system described in section 2.5 producing a large mean flow variation with turbulence superimposed on it.

The paper is structured as follow. Section 2 presents the experimental setup. Section 3 presents the control problem and the control strategies that will be exploited. Section 4 discusses the results and Section 5 gives some concluding remarks.

## 2  Experimental setup

The main purpose of this experiment is to highlight the feasibility of using advanced control algorithms within a simplified flow
configuration. Simplifications stand in the Reynolds number, the blade shape and the 2D section (no rotation and no transverse flow). This means that the flow characteristics (location of flow transition from laminar to turbulent, location of flow separation and thus aerodynamic loads) may differ from real applications. However, we show that even with such basic assumptions, the feedback control of the lift is possible and has sufficient robustness for potential usage in more realistic situations beyond 2D blade section assumptions.





In order to be self-content, we recall that the experimental closed-loop bench (see Figure 1 for pictures and Figure 2 for a functional scheme) is composed of a wind tunnel with its perturbation system (gust generator), a 2D aerodynamic blade profile equipped by micro-jets and aerodynamic lift and drag load sensors.

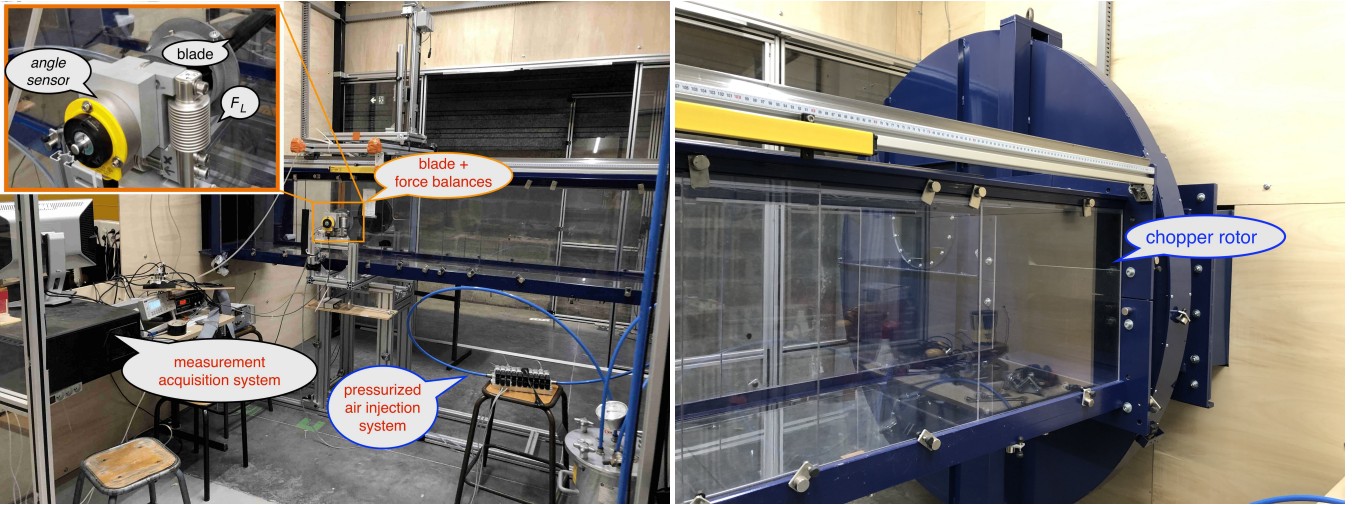

(a) Blade and its instrumentation & micro-jet system                    (b) Chopper rotor in the wind tunnel

**Figure 1.** Experimental setup within the wind tunnel.

## 2.1 Wind tunnel facility and gust generator

The LHEEA aerodynamic wind tunnel is a recirculating one. The test section has a cross-section of $500 \times 500 \text{mm}^2$ and a length
of $2300\,\text{mm}$ (Figure 2). The turbulence intensity of an undisturbed inflow in the wind tunnel is around $0.3\%$. In the present study, a grid is installed at the inlet of the test section to generate turbulent inflow with a turbulence intensity of $3\%$. This bypasses the laminar-to-turbulent transition occurring at low Reynolds numbers and low angles of attack (AoA) with respect to this blade geometry (see the linear part of the lift curve in Figure 3). The inlet of the test section is additionally equipped with a system which enables the generation of a sudden variation of the mean flow with turbulence superimposed on it (for
more details see Neunaber and Braud (2020)). This system is called "chopper" and consists of a rotating bar that cuts through the inlet of the test section (Figure 2).

## 2.2 Aerodynamic profile

A 2D blade section of type NACA $65_4 - 421$ with a chord length of $c = 9.6\,\text{cm}$ is installed in the test section of the wind tunnel[1]. It is a thick profile with two changes of the lift curve corresponding to a first boundary layer separation at the trailing

---

[1]Note that "2D blade section" here refers to a two-dimensional shape that is extruded in the third dimension so that the blade section spans the whole length of the wind tunnel.





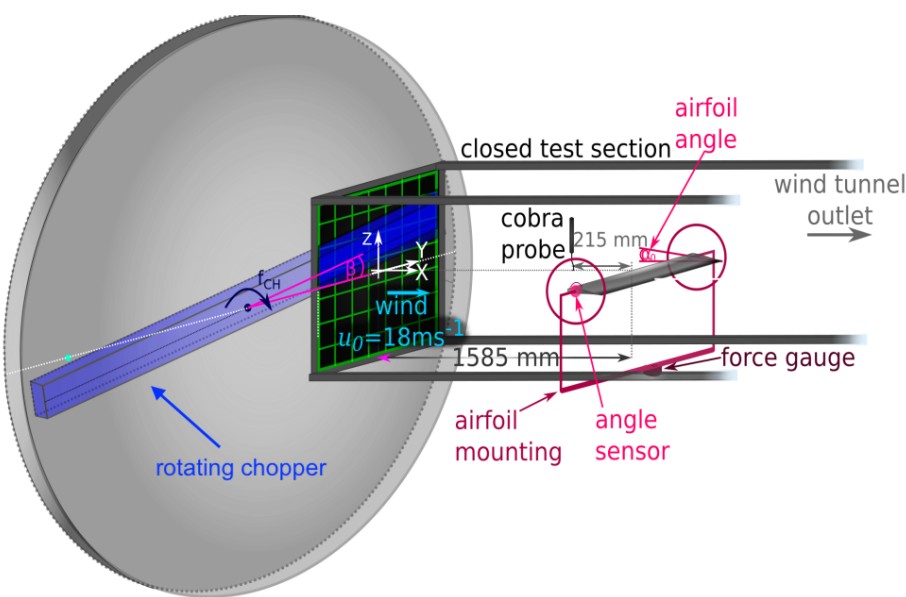

**Figure 2.** LHEEA aerodynamic wind tunnel including a gust generator, named "chopper", that creates lift perturbations and the system used for active flow control measurements: a tube with holes is installed at the airfoil's trailing edge and pressurized air is blown out of the holes.

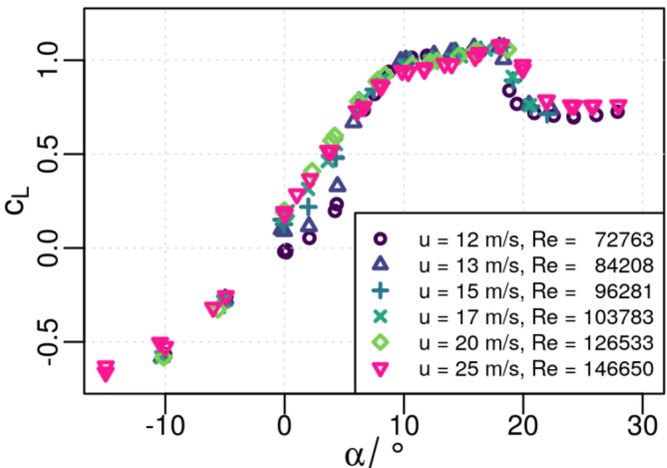

**Figure 3.** Experimentally measured lift curves, $C_L$, versus the angle of incidence, $\alpha$, of the airfoil in dependence of the Reynolds number.



edge of the profile and a second flow separation at the leading edge, indicating stall (see (Soulier et al. (2021)) for more details on the blade aerodynamics). In the present study, the angle of incidence is set to $\alpha = 20°$ that corresponds to the maximum of increase of the lift (see. Fig. 3) considering the additional micro-jet [2]. More investigations are needed to extend the present work to other angles of incidence.

## 2.3  Micro-jets

To control the flow around the airfoil, holes of $1\,\mathrm{mm}$ diameter with equidistant $8\,\mathrm{mm}$ spacing are placed at $1.92\,\mathrm{cm}$ from the airfoil's trailing edge, along the entire spanwise direction (Figure 2(c)). They are connected to a plenum chamber, itself fed with pressurized air at 6 bars. The air circuit is connected to solenoid valves that are acting as On/Off switches, so that pulsed micro-jets can be generated with a repetition rate of up to $300\,\mathrm{Hz}$.

The action of the micro-jets is physically limited to the injection of pressurized air of maximum 6 bars, thus defining the
range of the lift variations that can be compensated by the micro-jets actuator system. It is identified by a simple succession of valves opening and closing.

## 2.4  Lift and drag measurements

Two Z6FC3 HBM bending beam load cell sensors were used on each side of the blade support to measure the lift ($Y_1$, $Y_2$) and drag forces ($X_1$, $X_2$). They were calibrated *in situ* using calibrated weights from $0 - 5\,\mathrm{kg}$ in steps of $0.5\,\mathrm{kg}$.

## 2.5  Control hardware

The control is managed by a STM32 Nucleo board H743ZI2 allowing a 16-bit ADC acquisition as well as the possibility to monitor the signals in real-time on the computer. The lift force is measured by the force balances ($Y_1$, $Y_2$) and is acquired at a sampling rate of $20\,\mathrm{kHz}$ thanks to the Nucleo board. The signal is filtered using a fourth order Butterworth filter with a cut-off frequency of $20\,\mathrm{Hz}$. The control updates at $20\,\mathrm{kHz}$ and drives the valve at $200\,\mathrm{Hz}$ in response to the input from the force
balances.

---

[2]Some additional open-loop tests for $\alpha = \{0°, 10°, 20°\}$ have been performed to chose this value (see *e.g.* the Fig. 4 in Michel et al. (2024)). It has been shown that only a low range of the lift variation (or controlability margin of the lift) can be reached for $\alpha = 0°$ when the flow is still attached (*i.e.* maximum lift force gain $\Delta F_L = 2.5\,\mathrm{N}$). At $\alpha = 10°$, the controlability margin is three times higher ($\Delta F_L = 7\,\mathrm{N}$), but it is decreasing with the inlet pressure from $p = 1$ bar. An angle $\alpha = 20°$ is therefore chosen to operate the control algorithm, as the controlability margin is higher and linearly increasing according to the inlet pressure.





# 3 Control methodology

## 3.1 Problem statement

All along the paper, the control of the lift is performed thanks to a control loop that drives the pressurized air towards holes at the blade surface (Fig. 4), named micro-jets, which modifies the local pressure (that induces the lift), to track the lift reference.

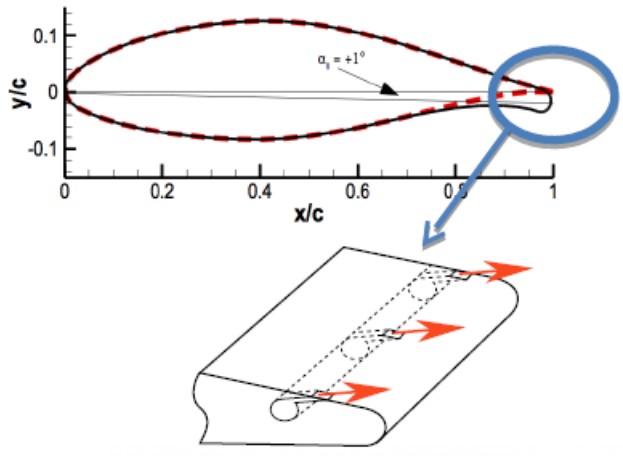

**Figure 4.** Mechanical configuration of micro-jets on the blade section.

In the sequel (Fig. 5), the control input of the system is denoted $u$, and the output that is controlled, denoted $y$, is the lift force. Considering the lift reference $y^*$, the goal of the closed-loop control is to ensure that the measured lift $y$ converges with "accuracy" to $y^*$.

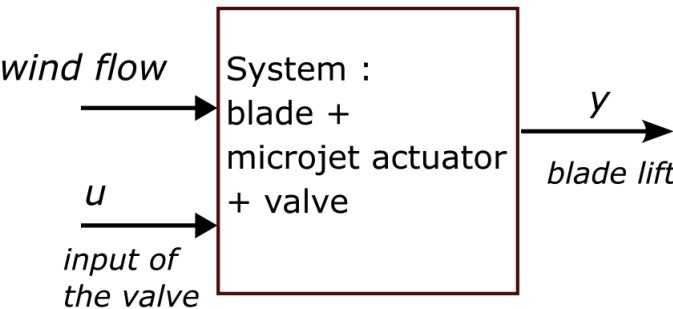

**Figure 5.** Scheme of the blade system including two inputs (the control $u$ & the wind flow) and the single output lift $y$.





The purpose of the study is to perform comparisons between controllers in order to evaluate performances such as for example time-response, tracking precision, delay to desaturate, taking into account the properties and the practical implementation
of each controller, regarding several operating conditions of the lift control system.

### 3.1.1 PID Robust Control (PID)

The lift variation in response to micro-jet actuation can be modelled as a second order system in first approximation (Brunton and Rowley (2010)). Such approximate model is highly dependent of particular aerodynamic operating condition, including specific inflow velocity, pitch, etc. One way to cope with several operating points as a whole is to consider such second order
models with uncertain parameters. A rather simple way to build uncertain models can be achieved by considering a finite set of relevant operating points at which specific models are identified, assuming that the true parameters to be in the convex set of the parameters obtained at these relevant operating points. Such modelling is known as polytopic modeling where state-space matrices can take infinitely many possible values within the set composed of all convex combinations of finitely many vertices. Robust control aims at assessing stability (and other performances) for all possible realisation of the system in the polytope.
Robust evaluations of performances are necessarily pessimistic compared to the true performance of the system at one specific operating condition. On the other hand, it gives guarantees of stability and other performances, at least, as long as the modeling assumptions hold true.

In this paper, we used control design results from (Conord and Peaucelle (2021)) that are implemented in the R-Romuloc toolbox (Peaucelle et al. (2014)). These results allow the design of state-feedback controllers. They rely on Lyapunov-type
methods and solve the design problem based on linear matrix inequality formulas solved by semi-definite programming tools. We identified state-space representation in controllable canonical form where the states are the error $e = y^* - y$ between the reference $y^*$ and the true lift measure $y$, and the time-derivative of this error. An artificial state as the integral of the error is added, hence, the state-space design provides exactly a PID controller of the type:

$$u = K_p e + K_i \int_0^t e(\tau)\,\mathrm{d}\tau + K_d \dot{e} \tag{1}$$

where $K_p, K_i, K_d$ are designed gains.

In order to evaluate the influence of the identified models on the performances of the closed-loop system, we performed the design procedure for three different choices of polytopes. The design is systematic assuming the learning of appropriate second order models is done and does not rely on tuning skills of some smart operator. The more operating points are considered, the more robust the controller shall be. Due to the upper defined pessimism, it may also have poor performance at specific operating
conditions. Poor performance can also come from discrepancies between true dynamics (which are not linear of order two) and the identified models.





### 3.1.2 Model-Free Control (MFC)

Full details on model-free control are given in (Fliess and Join (2013)). Its usefulness in many situations, including compensating severe non-linearities and time-varying reference signals, has been demonstrated (see *e.g.* Lafont et al. (2020) Park et al.
(2021)). The corresponding intelligent controllers are much easier to implement and to tune than standard PID controllers which are today the main tool in industrial control engineering (see, e.g., (Åström and Murray (2008))).

*The ultra-local model*

In the current application, the unknown description of the plant is restricted to a SISO (single-input single-output) system because the objective is to control only the lift $y$ (output) thanks to the actuator $u$ (input). The unknown description of the SISO
plant is replaced by an ultra-local first order model

$$\dot{e} = \dot{y}^* - \dot{y} = \dot{y}^* - (F + \beta u) \tag{2}$$

where: the control and output variables are respectively $u$ and $y$; the time-varying quantity $F$ is estimated online and subsumes the unknown internal structure and the external disturbances; the constant $\beta \in \mathbb{R}$ is chosen by the practitioner such that $\dot{y}$ and $\beta u$ are of the same magnitude. Therefore, $\beta$ does not need to be precisely estimated.
Equation (2) is only valid during a short time lapse that must be continuously updated: it implies that $F$ is estimated on-line through the knowledge of the control variable $u$ and the numerical differentiation of $y$. It is natural to consider firstly the ultra-local model (2) of the first order, for which, in the considered case, experimental results show that this particular order of the $F$ model gives results that are accurate enough regarding the present objective of the paper (track lift reference).

*Intelligent P controllers*

The control law reads as the *intelligent P controller*, or *i-P* controller

$$u = -\frac{F - \dot{y}^*}{\beta} + K_P e \tag{3}$$

where $\beta$ is a parameter and $K_P$ is a usual tuning gain that have to be set by the user.

The i-P controller (3) is compensating the poorly known term $F$. Controlling the system therefore boils down to the control of an elementary pure integrator. To numerically estimate the derivative of $y$, homogeneous semi-implicit differentiators (Michel
et al. (2021); Mojallizadeh et al. (2023)) have been used.

### 3.1.3 Adaptive Super Twisting control (AST)

Full details on adaptive super-twisting can be found in (Mirzaei et al. (2022)). The dynamics of the tracking error is assumed to be given by

$$\dot{e} = a + b\,u, \quad b \neq 0, \tag{4}$$

where $a$ and $b$ are unknown terms that are bounded in the operational domain; following the gained experience with MFC, it assumes that the relative degree[3] is equal to one. The objective in this work is to leverage the properties of the adaptive

---

[3]The relative degree corresponds to the minimum differentiability of the output $y$ before seeing the input $u$ (see Isidori (1985)).





super-twisting algorithm as a model-free control law, *i.e.*, without any knowledge of $a$ and $b$. From (Plestan and Taleb (2021)), the adaptive super-twisting controller is defined as

$$u = -k_1 |e|^{\frac{1}{2}} \operatorname{sgn}(e) + v$$
$$\dot{v} = -k_2 \operatorname{sgn}(e)$$

including the adaptive rules for the gains $k_1$ and $k_2$

$$\dot{k_1} = \begin{cases} \frac{\mu}{|\psi| + \epsilon_0} & \text{if} \quad |e| > \epsilon_0 \\ -k_1 & \text{if} \quad |e| \le \epsilon_0 \end{cases} \quad ; \dot{k_2} = \begin{cases} \frac{\mu}{2|e|^{\frac{1}{2}}} & \text{if} \quad |e| > \epsilon_0 \\ -k_2 & \text{if} \quad |e| \le \epsilon_0 \end{cases}$$

where $\psi = -\hat{e}, \quad \mu = \mu_0 \left( |k_1| \sqrt{|e|} + |\psi| + \int_0^t k_2 \operatorname{sgn}(e(\tau)) \mathrm{d}\tau \right), \quad (\epsilon_0, \mu_0) > 0$

and $\hat{e}$ is the numerical estimation of $\dot{e}$. The main advantages of the AST controller are:

–   the adaptive algorithm requires only limited information about the system modeling;

–   the adaptive algorithm is well-known for its robustness;

–   the adaptation of gains $k_1$ and $k_2$ helps reducing input energy consumption.

Note that these two approaches have different principles: the MFC can be considered as an extended classical control, based on an internal estimation of a ultra-local model, that approximates on-line the dynamic of the controlled system, whereas the AST is a high order sliding mode controller, whose gains are auto-adapted on-line and has conceptually a finite time

convergence instead of an asymptotic convergence for all other tested control laws.

Figure 6 depicts the corresponding closed-loop of the proposed control architecture including the wind and chopper perturbations.

### 3.2   Control law improvement

To deal with the physical limits of the micro-jet actuator, that may create uncontrollable situations and unexpected behavior

of the control algorithm in presence of strong perturbations of the lift, an anti-windup procedure (Tarbouriech et al. (2011)) is proposed to manage the integration part of the robust PID controller and the adaptive super-twisting algorithm when physical saturation occurs. In this study, only upper saturation is in effect due to the choice of particular experiment tests. Nevertheless, the actuator has upper and lower limitations due to the physical limitations of the experimental test-bench.

### 3.2.1   Discretization of the control

To implement the control laws into the STM board, discretized versions have been derived from the continuous versions presented in the previous section. Every control laws are "sampled" under basic Euler discretization strategies regarding the integrators parts: PID contains a single integrator and AST contains four integrators; these integrators were solved using for





example a numerical trapezoidal rule. However, contrarily to the PID (1), the MFC ultra-local model structure (2) does not contain integrators. Consequently, the sole discretization problem refers to the numerical time-derivative.

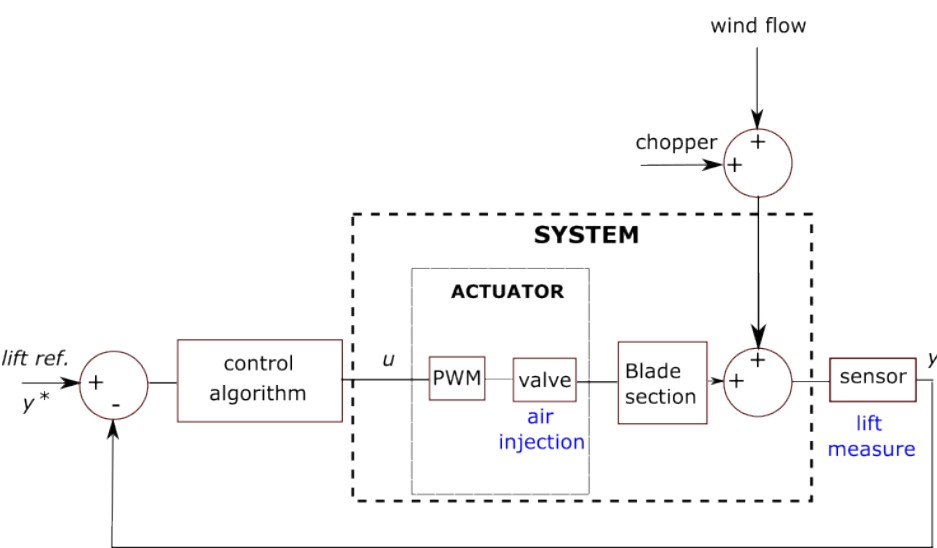

**Figure 6.** Closed-loop scheme.

Due to the binary nature of the solenoid valves, the "continuous evolution" of the control output is converted into variations of the duty-cycle signal of a square signal that drives the valves; the period of the PWM is set to $T_c = 200$ Hz. The quantification of the duty-cycle is a very important issue that may affect the quality of the tracking, even the stability of the overall control. Although the control algorithms provide "continuous" values, the conversion into the duty-cycle format requires to quantify the values of the control output since the practical implementation of the duty-cycle is incremental: in particular, the pressure

of 6 bars has been "swept" under a precision of 4000 points, meaning that the error of quantification of the pressure injected to the valve actuator is about $1.5\,10^{-3}$ bars. A high error of quantification gives less output resolution for the control to drive the pressure inducing strong oscillations of the measured lift. The choice of the precision is a compromise between 'minimal' tracking performances of the lift and the capabilities of the STM board (related to the maximum clock frequency) to increase the incremental precision of the duty-cycle since the control algorithm is updated at 20 kHz.

**3.2.2   Definition of the saturation**

The control is designed with respect to the saturation / physical limitation of the micro-jet actuator including an hysteresis and an anti-windup algorithm that interacts with the numerical integration schemes of the controller.

     Due to the physical limitation of the actuator, the output of the control is blocked / bounded when the control attempts to drive the lift outside the physical range of the admissible pressure of the valve actuator. The purpose of the anti-windup (AW)

algorithm (Tarbouriech et al. (2011)) is complementary to the control blocking and holds the value of the control by freezing





the integration part of the control algorithms to prevent from divergence or unexpected issue of the control. The anti-windup is tested in this work by considering the starting of the lift reference greater than the physical allowed pressure.

– In the PID case, the simplest solution is to freeze the integrator part during the saturation time.

– In the AST case, the simplest solution is also to freeze all the integrators during the saturation time, and especially the integration of the adaptive gains $K_1$ and $K_2$ since such adaptation is totally wrong while saturating.

– In the MFC case, blocking on the control law, the output of the control using a simple saturation is a solution to maintain the "learning effect" of the control and satisfy the input constraints.

## 4    Experimental results

### 4.1    Practical implementation

The PID control design requires at least to know a bit the dynamic of the system, whereas model-free based approaches (MFC and AST) require few information, like the relative degree and the sign of gain of the system.

– A robust PID controller is typically the simplest controller to implement within an embedded calculator and it can give excellent results for an 'unknown' dynamics, based on very rough modeling, but it is also very sensitive to changes and error of the modeling, which make the solution not efficient for a such application, as the unsteady aerodynamics on wind
215        turbine blades may vary significantly with atmospheric conditions. The recent advances in robust control design allows building a robust control based on rough polytopic modelling. This polytopic modelling induces also some implicit assumption with respect to the model validity domain with regards to our application. Three polytopic models have been considered from different operating conditions (inflow velocity variations and different blockage ratios), in order to synthesize three PID controller, whose robustness has been tested separately towards the different proposed modeling.
220        This approach requires an identification procedure to build the polytopes and the resulting synthesis of the robust control has been made thanks to a dedicated Matlab® toolbox. The more precise the polytopic approximation, the more effective the control, but this will require a lot of time and effort.

– The MFC is of the same complexity as the PID, including a prediction part that requires the estimation of a numerical time-derivative of $y$.

– The AST contains several integrators that manage the dynamic of the internal integrator and the dynamic of the gains.

Both model-free solutions are of interest because they do not need any prior modeling of the system making these solutions pretty well adapted to control fluid dynamics applications. The tuning of the MFC and AST has been made according to the gained experience (Michel et al. (2022, 2024)) and is consequently faster compared to the PID robust design (which requires a complete identification procedure).





## 4.2 Scenarios of operation

Several cases of operating conditions have been considered to compare model-based approach (robust PID control) and model-free based approach (MFC and AST) in terms of performances (SSE and STD index defined later in p. 14) as well as the energy of the control signal, the time responses and the desaturation time. The efficiency of the lift tracking is evaluated for several scenarios that illustrate different operating conditions, defined by different inflow velocities and different fixed positions of the chopper in the test section, for which the characteristics are described below and are summarized in the Table 1. The chopper induces different perturbation levels defined as the ratio in percentage $S_p = \frac{S_{bar}}{S} \times 100$, with $S_{bar}$ the chopper surface area introduced in the test section and $S$ the surface area of the test section such as $S_p = \{0, 0.6, 2.5\}\%$. Due to the difference of dynamic between the chopper displacement and aerodynamic phenomena including the micro-jet feedback loop, in this paper, the chopper is maintained at a fixed position for which its displacement is considered as instantaneous.

The experiments are conducted considering a constant inflow velocity of $20\,\mathrm{m.s^{-1}}$, except for the scenario 1 which considers an inflow velocity of $19\,\mathrm{m.s^{-1}}$, measured with a Pitot tube in front of the airfoil in the undisturbed flow (before the chopper), and an angle of incidence of the 2D blade section of $20°$. The chopper, when introduced slightly in the test section, reduces the mean inflow velocity and adds turbulence. The chopper is used to evaluate the robustness of the controllers under perturbations of the lift and quantified using the ratio between the chopper surface area and the test section area as introduced in the previous Section 2.1. Reported in percentage, it represents the blockage coefficient created by the chopper; the time duration during which the chopper is introduced in the test section is indicated together with the blockage coefficient in Tab 1.

**Scenario 1:** This starting scenario considers the simplest case where the inflow velocity is set to a constant low value of $19\,\mathrm{m.s^{-1}}$ and no perturbation is introduced.

**Scenario 2:** The inflow velocity is set to $20\,\mathrm{m.s^{-1}}$ and the chopper is introduced at $t = 30$ s to disturb the air flow (fixed at 0.6 %).

**Scenario 3:** The inflow velocity is set to $20\,\mathrm{m.s^{-1}}$ and the chopper is introduced at $t = 30$ s to disturb the air flow (fixed at 2.5 %).

**Scenario 4:** The inflow velocity is set to $20\,\mathrm{m.s^{-1}}$, then changed to $21\,\mathrm{m.s^{-1}}$ at $t = 30$ s. The chopper is introduced at $t = 20$ s to disturb the air flow (fixed at 0.6 %).

**Scenario 5:** The inflow velocity is set to $20\,\mathrm{m.s^{-1}}$, then changed to $21\,\mathrm{m.s^{-1}}$ at $t = 30$ s. The chopper is introduced at $t = 20$ s to disturb the air flow (fixed at 2.5 %).

**Scenario 6:** The inflow velocity is set to $20\,\mathrm{m.s^{-1}}$. The chopper is introduced at $t = 30$ s to disturb the air flow (fixed at 2.5 %) and removed at $t = 80$ s.

To illustrate the operation of the saturation mode, a higher output reference than the maximum reachable lift is firstly considered in order to saturate the micro-jet actuator, then a piece-wise constant reference is applied to track the lift.





| Sc. # | Inflow velocity | Blockage ratio |
|---|---|---|
| 1 | 19 m.s$^{-1}$ constant | – |
| 2 | 20 m.s$^{-1}$ constant | 0.6 % starting at 30 sec |
| 3 | 20 m.s$^{-1}$ constant | 2.5 % starting at 30 sec |
| 4 | 20 m.s$^{-1}$then 21 m.s$^{-1}$ starting at 30 sec | 0.6 % starting at 20 sec |
| 5 | 20 m.s$^{-1}$ then 21 m.s$^{-1}$ starting at 30 sec | 2.5 % starting at 20 sec |
| 6 | 20 m.s$^{-1}$ constant | 2.5 % @30s and remove @80s |

**Table 1.** Overview of scenarios of operation.

### 4.3 Setup of the controllers

Table 2 summarizes parameters of controller that have been used for each scenario. In particular, concerning the PID control, the (a), (b) and (c) controllers have been synthesized based on three polytopes that combine several operating conditions, that are summarized in Table 3.

| type | $Kp$ | $\beta$ | $\epsilon_0$ | $\mu_0$ | $P_{Kd}$ | $P_{Ki}$ | $P_{Kp}$ |
|---|---|---|---|---|---|---|---|
| MFC | 0.0002 | 0.0002 | | | | | |
| AST | | | 20 | 1.5 | | | |
| PID(a) | | | | | $1.37 \cdot 10^{-6}$ | 2.498 | $1.96 \cdot 10^{-4}$ |
| PID(b) | | | | | $-1.3 \cdot 10^{-7}$ | 5.675 | $1.906 \cdot 10^{-4}$ |
| PID(c) | | | | | $-2.4 \cdot 10^{-7}$ | 5.975 | $1.901 \cdot 10^{-4}$ |

**Table 2.** Parameters of the controllers

| | inflow velocity 19 m.s$^{-1}$ | | | inflow velocity 20 m.s$^{-1}$ | | | inflow velocity 21 m.s$^{-1}$ | | |
|---|---|---|---|---|---|---|---|---|---|
| polytope | 0 % | 0.6 % | 2.5 % | 0 % | 0.6 % | 2.5 % | 0 % | 0.6 % | 2.5 % |
| (a) 8 matrices | X | X | X | X | X | X | X | X | |
| (b) 3 matrices | X | X | X | | | | | | |
| (c) 3 matrices | | X | | X | | X | | | |

**Table 3.** Polytope definitions of the PID controllers.

Remark that, in the case of the polytope (a), if the 2.5 % for the inflow velocity of 21 m.s$^{-1}$ is also included, then there is no solution to the robust control problem.

### 4.4 Results and discussion

In this section, the experimental results are presented considering firstly no actuator saturation during the lift tracking, and then, with actuator saturation.





### 4.4.1 Analysis of performances in case of no control saturation

Throughout this sub-section, it has been verified that the measured lift does not saturate, meaning, that the evolution of the input $u$ is not limited by the AW algorithm.

The performances of the controllers with respect to the tracking error are evaluated thanks to the usual performances index
275 criteria including the *sum of square error* (SSE), *standard deviation* (STD) and *variance of the control input* (VarU), which informs about the control effort of each controller [4]. Each index is averaged over the scenarios of operation and the global comparison is presented into an histogram in Fig. 7.

To illustrate the operation of each control, the Scenario #2 has been selected. Figures 8, 9, 10, 11 and 12, illustrate the tracking of the instantaneous lift for each MFC, AST, PID(a), PID(b) and PID(c) controller, according to the evolution of the
280 corresponding duty-cycle.

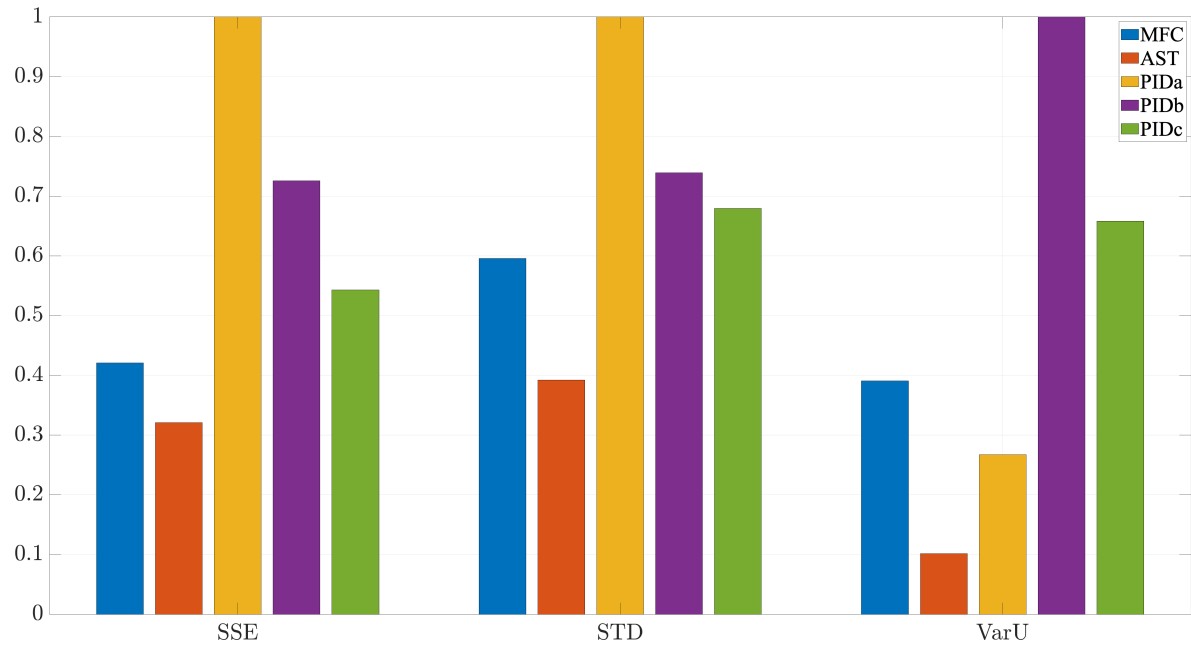

**Figure 7.** Normalized performances SSE, STD and VarU of the control laws averaged over scenarios for each controller.

The model-free based approaches require very few information about the dynamical system to control. In particular, the lack of information about the system uses learning properties or adaptive properties of such controller that has to "guess" the behavior of the system. The AST offers globally better performances over all scenarios than MFC controller due to adaptive

---

[4]The SSE of the error is given by $\mathfrak{E}_y = \sum_k (e)^2$ and the variance of $u$ is given by $Var_u = \sum_i |u_{k+1} - u_k|$.





integrator and adaptive gains associated to a sliding mode controller, that smooth the response according to high frequency
variations of the lift dynamics. The performances are however very similar in Sc. #2 depicted in Fig. 8 and 9. On the other
hand, the MFC contains an anticipating action (via numerical derivation) instead of the adaptive action, which makes this
controller more reactive to small aerodynamic perturbations.

The comparison of the tracking between the three PID controllers shows that the particular controller, associated to the
polytope (c), shows a good tracking of the lift in the case of the perturbed wind flow including a inflow velocity change, meaning
that this particular polytopic model matches the best the overall dynamics, whereas the other polytope based controllers give
worse performances. Despite the strong difficulty to model the dynamic of the lift, one can assume that a very particular choice
of the operating conditions to build the polytopic model could give interesting results. Note that the identification process to
tune the robust PID control is difficult to maintain rigorous operating conditions in the wind tunnel considering this additional
turbulence generated by the presence of the chopper. The identification has been performed over an average of several step
responses of the measured lift, for which it is assumed, in closed-loop dynamics, that the eigenvalues of the control loop
design is done in order to consider the dynamic of the micro-jet action continuous (averaged approximation). We yet notice a
posteriori that the identification assumptions that the models depend of operating conditions and not on the average duty-cycle
(the effective actuation force) seems erroneous: this reflects very different dynamics depending on the value of the reference
lift, for which a closer identification would improve the models.

Focusing on the static response of each controller around 15 sec, Figs. 13 and 14 highlight the behavior of each controller
for the Sc. #2 and #5 respectively. In both cases, MFC and AST show a smooth responses, while the responses of the PID
controllers remain harmonic, inducing a worse rejection of the aerodynamic perturbation. Globally, the nonlinear properties of
AST and MFC allow a better reduction of the aerodynamic perturbations whereas PID control is limited to rejects disturbances
and tends to amplify oscillations (Fig. 13). Nevertheless, if the perturbation, induced by the chopper and variations of the mean
inflow velocity, increases, the rejection becomes less efficient for the nonlinear control AST and MFC (Fig. 14) but remains
better than PID. Moreover, as observed previously in Sc. #2, Fig. 10, 11 and 12 highlight differences of the tracking efficiency
according the level of the lift reference, hence, showing the limitation of the considered robust PID control.

Remark that in Fig. 9, the time needed for the AST controller to converge to the reference is slower due to the adaptation
of the gains; this convergence issue is not general and depends strongly on the operation conditions. Due to the internal
anticipation of the MFC structure, Fig. 8 shows good convergence to the reference, which is very similar to the behavior of the
PID controllers.

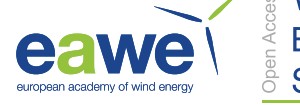

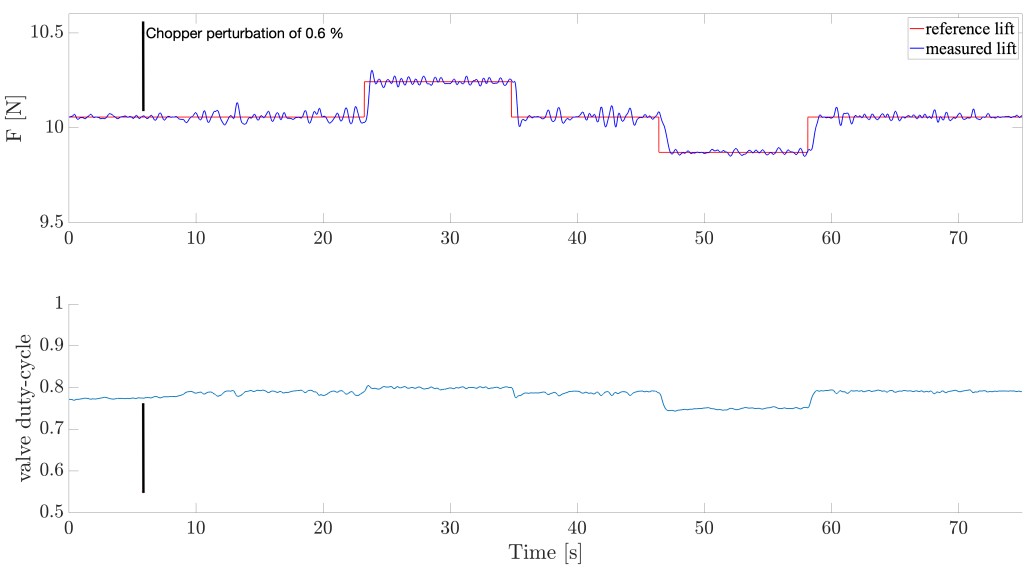

**Figure 8.** Time Evolution of the lift controlled by the MFC algorithm and associated duty-cycle.

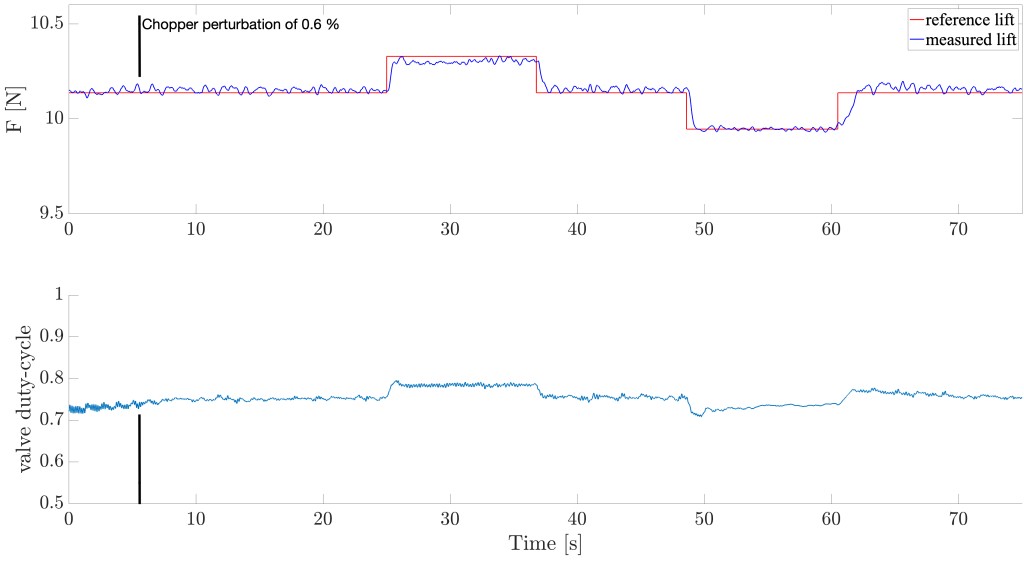

**Figure 9.** Time Evolution of the lift controlled by the AST algorithm and associated duty-cycle.





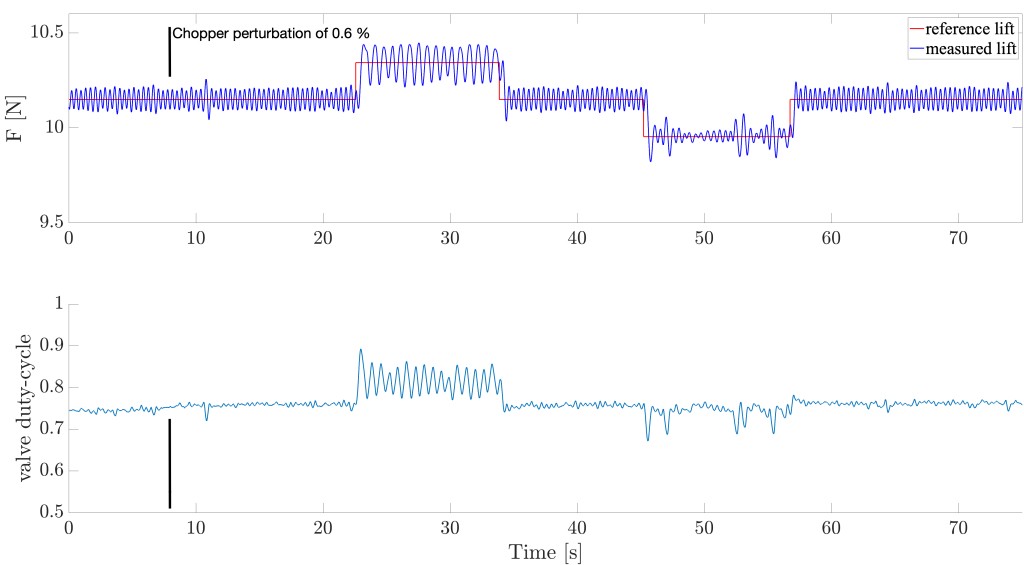

**Figure 10.** Time Evolution of the lift controlled by the PID(a) algorithm and associated duty-cycle.

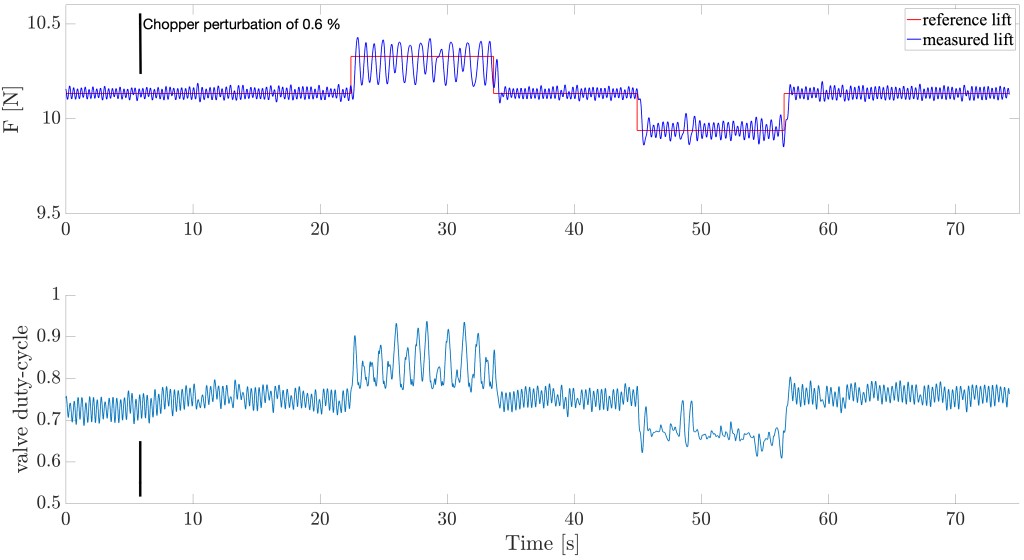

**Figure 11.** Time Evolution of the lift controlled by the PID(b) algorithm and associated duty-cycle.





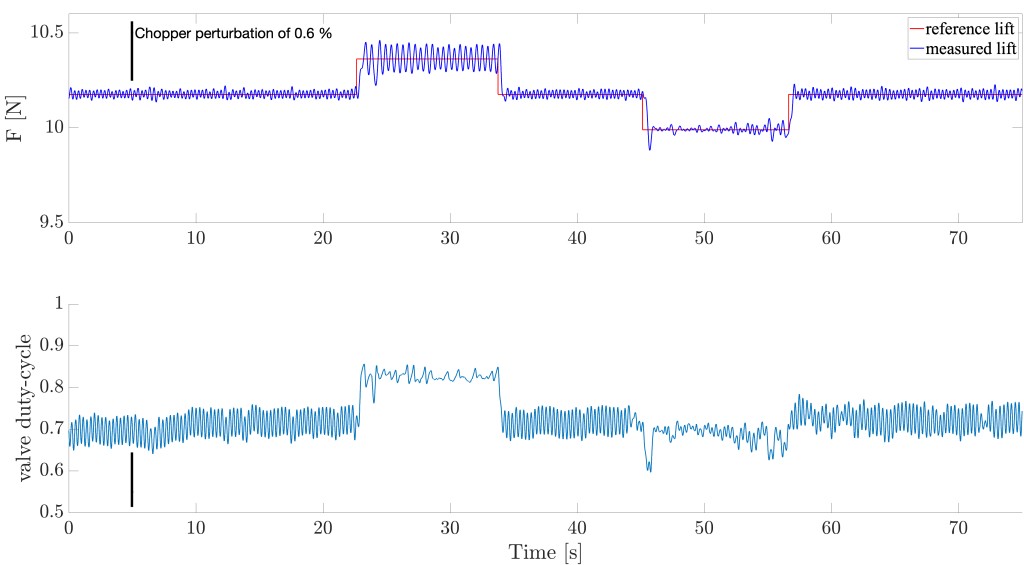

**Figure 12.** Time Evolution of the lift controlled by the PID(c) algorithm and associated duty-cycle.

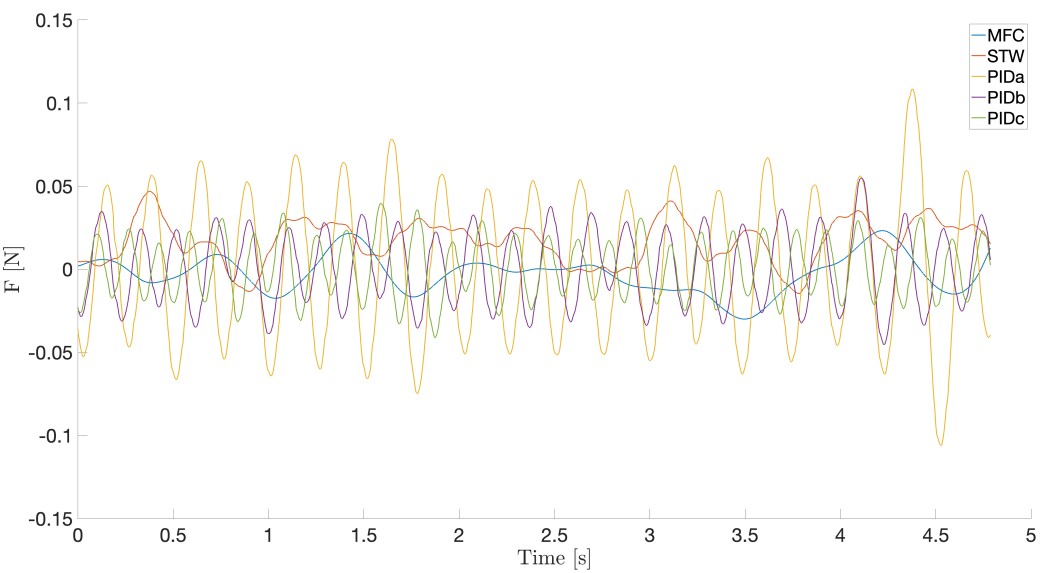

**Figure 13.** Sc. #2 - Comparison of the Time Evolution of the lift with each controller.



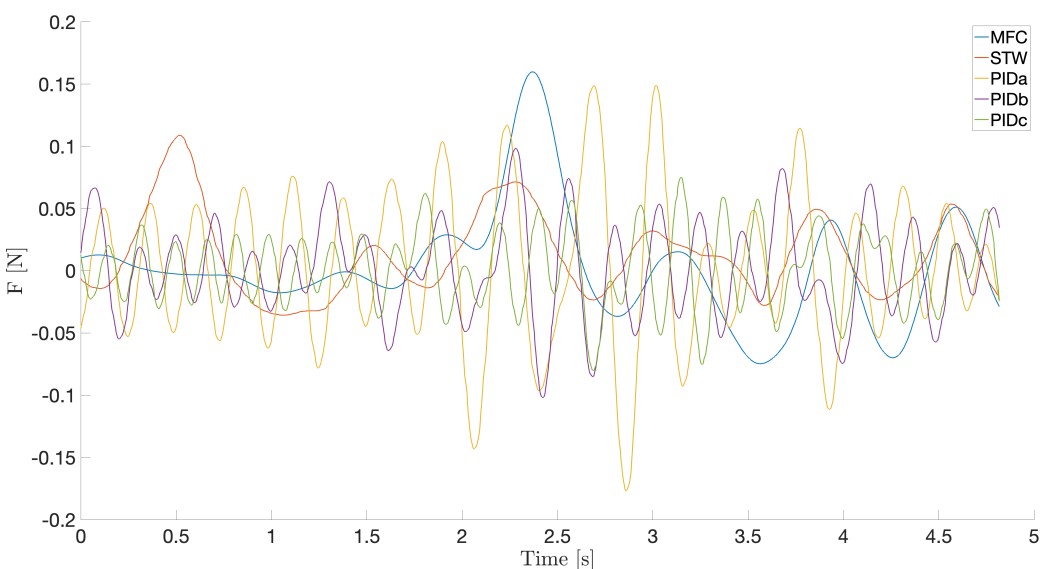

**Figure 14.** Sc. #5 - Comparison of the Time Evolution of the lift with each controller.

### 4.4.2 Analysis of performances in case of control saturation

The problem of saturation comes from the presence of integrator inside the controllers, hence introducing an anti-windup algorithm to prevent the integrators from diverging (that induces in this case a saturation of the output $y$). As the MFC controller does not contain a numerical integrator, the problem of the desaturation using an AW algorithm concerns only the AST and PID controllers.

In the case of the Scenario #2, Figs 15, 16, 17 and 18, illustrate the tracking of the instantaneous lift for respectively the AST, PID(a), PID(b) and PID(c) controllers during the saturation mode, involving the anti-windup (AW) algorithm, at the beginning of the control operation.

The easiest AW algorithm is typically applied for the PID control, for which a single integrator needs to be frozen when saturation occurs. At the opposite, the more complex one is the AST, for which it is required to freeze four integrators ($v$, $\mu$, $k_1$ and $k_2$) using a particular sequencing with respect to the gains management, which makes the tuning of its AW more difficult. The desaturation depends mainly on the integrators and gains, hence, the AST desaturation is slower. This is due to the time needed to re-adapt the gains (in the considered PID, the gains are fixed). Remark that in Fig. 15, the behavior until 10 sec corresponds to the initialization sequence of $k_1$ and $k_2$ integrators. In order to highlight the initial adaptive time, both gains are initialized at very small values.



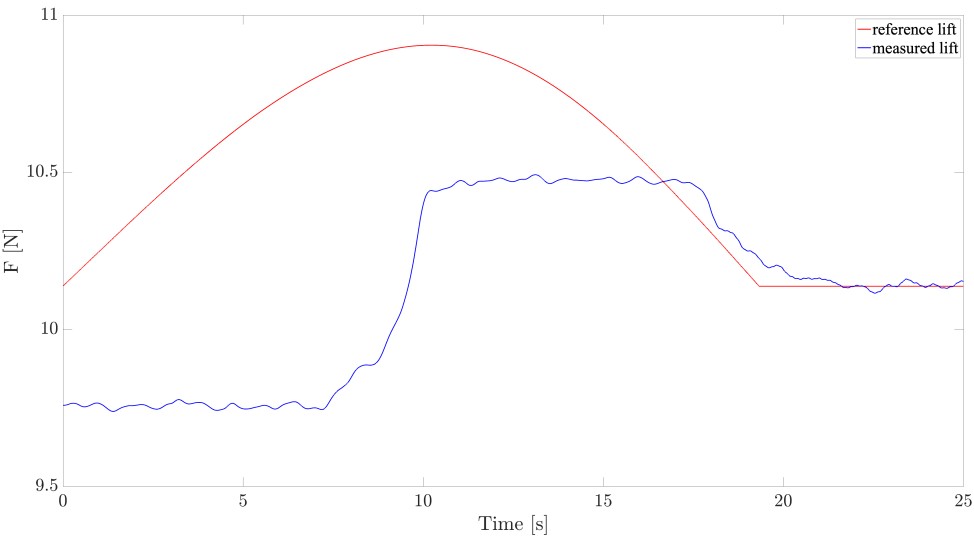

**Figure 15.** Time Evolution of the lift controlled by the AST algorithm under saturation of the reference.


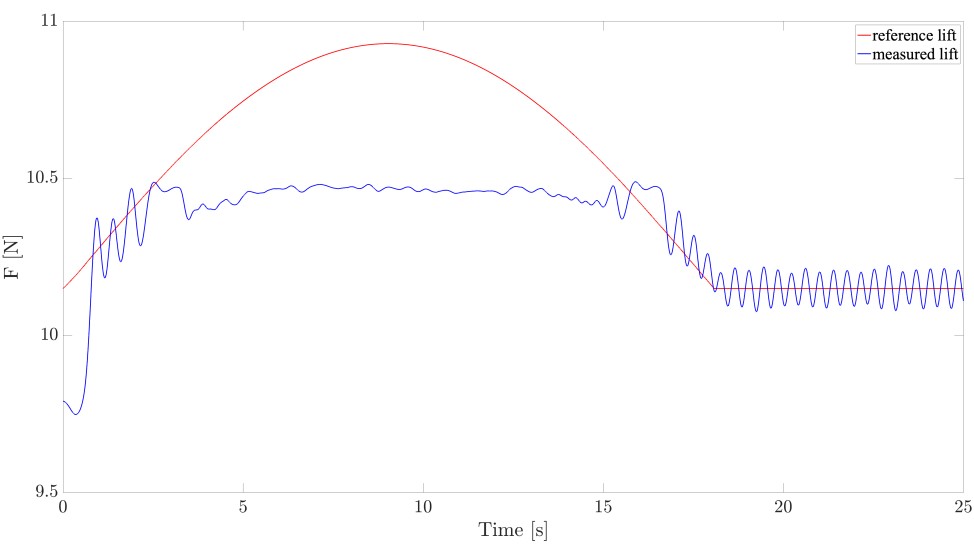

**Figure 16.** Time Evolution of the lift controlled by the PID(a) algorithm under saturation of the reference.

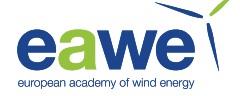
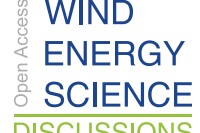


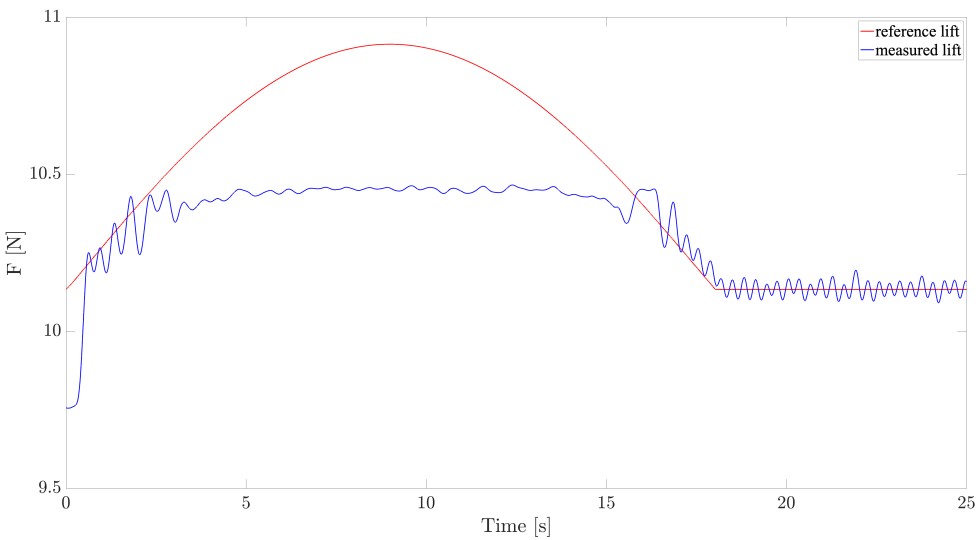

**Figure 17.** Time Evolution of the lift controlled by the PID(b) algorithm under saturation of the reference.

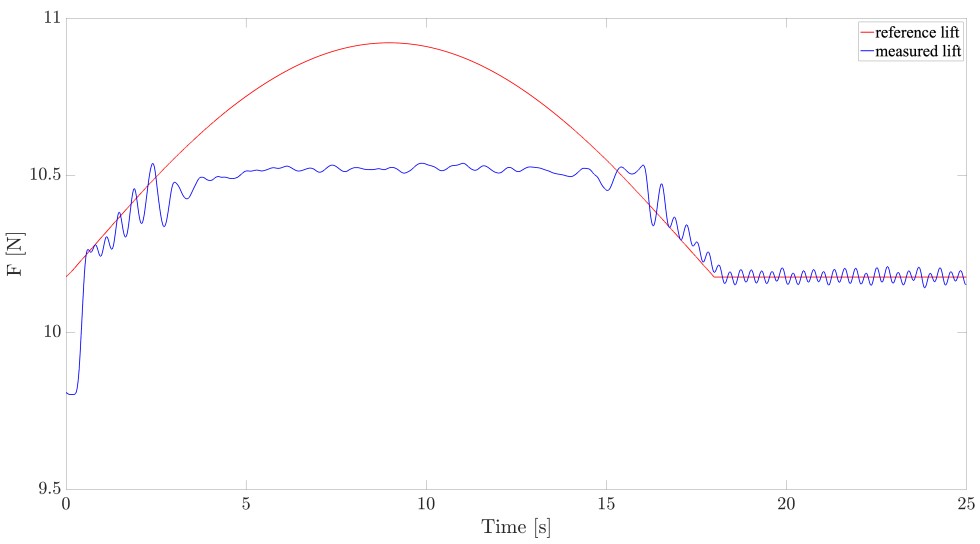

**Figure 18.** Time Evolution of the lift controlled by the PID(c) algorithm under saturation of the reference.



## 5 Conclusions

This work provides some feedback from the gained experience about the practical implementation of dedicated control laws that
are suitable for the aerodynamic lift control based on active flow control. The investigations have been carried out considering
on one hand, a model-based control, such as robust PID control, for which difficulties of prior modelling have been pointed
out in the experimental aerodynamic wind turbine framework at the laboratory scale, and on the other hand, model-free based
controllers, that do not require complete prior modelling and for which the tuning appears more flexible. The results highlighted
the fact that even with simple control laws, it is possible to control very complex systems.

Regarding the controllers design:

– The performances of the robust PID control is linked to the quality of the identified linear models used to define the
polytopes. Meanwhile, adding AW features is straightforward and very simple to tune.

– MFC requires only few information, and gives interesting tracking performances even if it not well tuned. It does not
need any AW.

– AST requires also only few information, and shows very good tracking performances outside the saturation mode. The
AW has to be improved according to the adaptive procedure of the gains.

Comparing all controllers performances and the design & development time, MFC appears as a good compromise in case of
saturation but AST provides slightly smoother responses in case of no saturation.

Perspectives include a complementary focus on the lift control based on the drive of the pitch angle of the blade in order to
cope with a better lift range, including the utilization of aerodynamic sensors such as remote wall pressure sensors or remote
flow separation sensors. Further investigations will deal with the coupling of both *local* (using the microjets) and *global* (using
the pitch) control approaches in order to better reject lift fluctuations, as well as, to investigate deep learning approaches.
Moreover, it appears that coupling both the micro-jet-based control and the pitch-based control would be a more advantageous
solution regarding local and global lift tracking.

*Code and data availability.* All data (dataset and Matlab scripts to plot all figures) are available on this repository:
https://zenodo.org/doi/10.5281/zenodo.10617750.

*Author contributions.* L. Michel carried out the conceptualization, the methodology, investigations, the software design and the preparation
of the full paper. C. Braud, J.-P. Barbot and F. Plestan carried out the conceptualization, the methodology, investigations and the preparation
of the full paper. In addition, C. Braud provided the experimental resssources and C. Braud and F. Plestan provided the funding acquisition.
D. Peaucelle provided knowledge about robust PID design tools as well as a ready to use code for designing the control gains for given
polytopes, and participated to the review of the paper. X. Boucher participated to the software design and the writing of the paper.



*Competing interests.* The authors declare that they have no conflict of interest.

*Acknowledgements.* The authors thank Michel Fliess and Cédric Join for they fruitful advices with respect to the model-free control algorithm implementation, which has been the starting point of our previous investigations of this study. The authors thank Dr. Pierre Molinaro for the design of the electronic control board in the experimental setup. This work was partially supported by ANR (*Agence Nationale de la Recherche*) with project CREATIF ANR-20-CE05-0039, by WEAMEC(West Atlantic Marine Energy Community) cluster with the projects ASAPe and FOWTBLADE, and by CARNOT MERS with project GOWIBA. Jean-Pierre Barbot is supported with (*Region Pays de la Loire*) Connect Talent GENYDROGENE project.



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
