# Peer review of "Comparison of different feedback controllers on an airfoil benchmark"

_Wind Energy Science, 2024_

## Referee Comment (RC2)

[referee-annotated manuscript omitted]

---

## Author Response (AR1)

We thank the reviewers for their time and attention to the manuscript and for the insightful comments which we feel have helped us improve the manuscript.
Changes made to the manuscript are clearly visible in the supplementary file where new text passages are marked in blue.

Responses to the reviewers are organized as follows:
- Initial reviewer's questions are in blue
- Author's comments are in black
- Changes made in the document are in bold black

**Review R1**

**Q1.1 : I think the paper title is weak because of the "first three words: "Some comments on." Can you just leave these words out?**

We thank the reviewer for the comment and suggestion. We strengthened the title while concentrating on the main feature of the paper as follows:
**"Comparison of different feedback controllers on an airfoil benchmark"**

**Q1.2 : There has been done quite a lot of research on local blade section control. Please extend your introduction by discussing and referencing these works and positioning your work in relation to what has already been done.**

We thank the reviewer for these remarks, the literature on the local control of wind turbine blade has been extended including more references and review articles on the subject as follows :
**« More recently, different control technologies for wind energy applications were reviewed in Aubrun *et al* (2017). Along with the development of AFC devices and open-loop tests came the development of closed-loop tests using advanced controllers with the early work of Allan *et al* (2000) using a model-based approach. A comprehensive review of control strategies dedicated to gust alleviation problems using active flow control is presented in Williams (2018). Feedback and feed-forward structures based on system identification have been investigated for active load reduction in the context of a controlled wind turbine blade (see e.g. Barlas (2008), Balas (2013), Jaunet et al (2018), Peaucelle et al (2019), Bartholomay et al (2021)). While some model-free approaches were explored in the same control objective by Becker et al (2005), Michel et al (2022, 2024), based on different well established modeling techniques Scheinker (2024); Fliess and Join (2021); Shtessel et al. (2023). »**

According to reviewer's comments, we also strengthened the positioning of our work by the addition of the following sentence in the introduction :
**« However, none of the control algorithms were compared on the same airfoil benchmark. It is well-known that airfoils may exhibit very different phenomena due to different shapes, Reynolds numbers or turbulent intensities etc ... (see e.g. Mc Cullough and Gault (1951), Gault (1957), ) that are still investigated (see e.g. Brunner et al (2021), Braud et al (2024)). In the present work different control strategies, model-based and model-free types, were investigated on the same airfoil configuration which serves as a benchmark to highlight pros and cons with respect to different criteria. »**

**Q1.3 :** **The introduction does not state anything about the outcome of this research, please include this.**

We thank the reviewer for this comment and suggestion. We modified the end of the introduction as follows:

**"The purpose of this study is to evaluate the performances of some selected feedback control strategies under different operating conditions of an experimental airfoil bench. The main goal is to alleviate aerodynamic load fluctuations and is tested with respect to large mean flow variations with turbulence superposed to it. The experimental setup is by itself a contribution as it can serve in the future to test more control laws in different configurations which are both simple and realistic compared to industrial applications. The second, yet main contribution, is to design and test three types of controllers: the robust Proportional-Integral-Derivative or PID (Peaucelle et al (2021), the Model Free Control or MFC (Fliess and Join (2013), Fliess and Join (2021)), and the Adaptive Super-Twisting algorithm or AST Shtessel et al (2023). The tests allow the characterization of operating domains for each control law regarding criteria like the nominal lift responses, the rejection of high frequency fluctuations, and the robustness with respect to modifications on the dynamics due to changes of the air flow characteristics. The outcome of the study is a comprehensive exposure of the pros and cons of each feedback control approach, not only for the produced performances for the load control itself, but also in terms of needed efforts for the design and the implementation of these controls."**

**Q1.4 :** **The introduction should clearly state the contributions. Is it that you evaluated three different controller types on an already existing test bench?**

We thank comments from the reviewer and have added explicitly what are the main contributions of the paper. See above the rewritten last section of the introduction. The contributions are: the experimental setup itself which may serve for other control law comparisons; the exposure of how three controls were designed for the benchmark; the descriptions of advantages and disadvantages we have put in evidence.

**Q1.5 :** **The introduction should make clear why you chose these three controllers. Now feels a bit random. Also, these controllers are not well described in the introduction.**

Again, we thank the reviewer for the suggestion for improving the Introduction. It has been revised as follows:

**"In this work, three control laws are being investigated. They have been chosen based on the experience of the automatic control collaborators of this study and, rather than investigating all possible solutions at their hand, we specifically selected those control strategies that are suitable for cases where there is little precise knowledge on the system to control and limited algorithmic complexity on implementation level. The three control laws are: (a) a robust PID controller (Conord and Peaucelle (2021)) which has the simplicity of the classical basic PID while providing potentially robustness and close to equilibrium perturbation rejection performances; (b) a model-free control (Fliess and Join (2013, 2021)) which requires little online tuning; (c) an adaptive sliding-mode control based on the super-twisting algorithm (Shtessel et al. (2023)) which also requires little knowledge about the model and has interesting finite-time convergence properties.**

**Q1.6 :** **It is not clear to me whether you can individually open/close each of the holes in the blade (you say something about multiple solenoid valves?). Maybe indicate and make it clear using a figure?**

With the actual experimental set-up, we cannot individually open/close each of the holes in the blade. We thank the reviewer for this remark, we reformulated the description of the control technology to make it clearer in the document as follows :

**« The plenum chamber is an hollow tube placed along and inside the blade, in the spanwise direction, tangent to the airfoil surface, with holes on it. Jets are coming out of these holes when hollow tube ends are connected to the air circuit. This prevents the individual control of jets, however this ensures the jet amplitude homogeneity in the spanwise direction. The air circuit is connected to solenoid valves that switch On/Off simultaneously, using a single control law so that synchronous pulsed micro-jets can be generated with a repetition rate of up to 300 Hz.»**

**Q1.7 :** **Section 3 is not introduced, you directly begin with 3.1.**

Thank you for noticing, a transition sentence has been added as follows:

**"In this section, the control problem is presented as well as the three control algorithms (PID, MFC and AST). The implementation method is also described with in particular an introduction to the anti-windup procedure, the discretization method and the saturation method."**

**Q1.8 : So you consider a SISO system and no actuation of individual holes? I.e.: Output lift force (y), that needs to be controlled (follow a reference lift, y*) by a pressure input (u)? Be more clear on this.**

We thank the reviewer for this remark, we believe it is now clearer in the document with the answer provided in question 1.6.

**Q1.9 : The performance that you can attain with each of the controller types is very dependent on the calibration and implementation of the controller. I think it is hard to conclude that one of the controllers is superior with respect to the other controllers, using your current approach. That is, how do you know for sure that you cannot attain the performance levels of the other controllers, simply by a more optimal calibration of the gains? Please provide more discussion and insights on this point. What would your approach be to get the most optimal performance of all the controller so that you can draw these conclusions?**

We agree with the reviewer that the tuning of the control parameters may have a significant impact on the closed-loop. As far as the study could go, we provide some values for these gains that are reasonable considering the adopted design strategies. We do not claim these values have any optimality; it may well be that we tune the parameters to have similar results with two or three of such controllers. But our goal was not to tune controllers with the objective to do the same as another given one.

To take into account your valuable questions in a fair manner, the conclusion is modified as follows:

**"This work provides comprehensive knowledge from the gained experience about the practical design and implementation of some feedback control laws that succeed in performing aerodynamic lift control with active flow actuators. The investigations lead us to conceive an appropriate test-bench focusing on the lift control problem. Three control strategies have been selected, the parameters of the controllers were designed and implementation has been carried out on the test-bench. In summary the conclusions for each control are the following. The model-based strategy for the design of robust PID control has the advantage of being rather systematic but is highly dependent on prior model identification. As it assumes linear representation of the plant it is expected to perform mainly when the system is close to the set point. The two other control strategies are model-free (or assuming basic properties on the plant) but require hand tuning which may not be systematic. It was revealed to be rather simple in the MFC case, and did not need a posteriori to build some Anti-Windup strategy to cope with saturation issues. The features of the adaptive super-twisting control revealed rather smooth time-responses. Comparing all tuned controllers in terms of close-loop performances but also the design and development time, MFC appears as a good compromise in case of saturation while AST provides slightly smoother responses. We are conscious that the conclusions in terms of performances may differ when applying other values on the control parameters. We claim not that the values are unique nor optimal. It may well be that the hierarchy of results changes for other versions of these same controls. One of the perspectives for the Automatic Control colleagues involved in the project, is to establish mathematical tools to tune control parameters such that the controllers, at least locally, provide similar results. But this is out of the scope of the present work."**

**Q1.10 : Subscripts that are not indices should be nonitalic, e.g., $K_\mathrm{p}$**

Thank you for noticing the typo, it has been corrected everywhere in the paper.

**Q1.11 : Captions underneath figures need to be more extensive, so that figures are interpretable independent from the main text.**

Thank you for this comment, captions of the figures have been extensively reviewed to make them interpretable independent from the main text.

**End of review #1**

**Review R2**

**Q2.1 : From line 11 to 14 are just repeating exactly the same sentence in abstract. Please consider modifying it.**

We would like to thanks the reviewer for this remark which serves to improved our abstract as follows:

"**The present paper proposes a comparison of three well established controllers: a robust PID controller (Peaucelle (2021)), a model-free control (Fliess and Join (2013, 2021)) and an adaptive sliding-mode control based on the super-twisting algorithm (Shtessel et al. (2023)). The benchmark considered is an airfoil section equipped with trailing edge jets, load sensors and a perturbation system. The objective is to track the lift command under external wind perturbations. Outcomes of this work are comparison of performances for three control laws that are suitable when little knowledge is known from the physics. This study not only quantifies performance in terms of load control, but also in the needed implementation effort.**"

**Q2.2 :** **Line 23,"(Bartholomay et al. (2021)) developed a feed-forward controller ...". The parentheses are not needed. The same applies to all the citation in the paper.**

Thank you for noticing, the document has been reviewed to remove parentheses when not needed.

**Q2.3 :** **At line 36,"PID", this term appears in the first time, you need to spell the whole words.**

The definition of PID has been added the first time it appears in the document by the following: **"Proportional-Integral-Derivative"**

**Q2.4 :** **The information provided by Figure 5 is very limited, either adding more information to the figure or removing it.**

Thank you for this remark. Captions of all figures were reviewed.

**Q2.5 :** **There are many occurrences of "thanks to" in the paper, which makes the sentence to be seen as informal or conversational. It does not fit with a technical paper. Please consider modifying it.**

Thank you again for this remark. Sentences with "thanks to" have been rephrased.

**Q2.6 :** **what is "The ultra-local model". Please describe it or use any citation.**

A citation now appears in the paper as follows:

**"that approximates very locally the overall dynamics of the system, (see \cite{FliessJoin_2013})"**

**Q2.7 :** **Please explain in more detailed regarding how to estimate "F" used in Eq. (2) since this is very important parameter.**

F is estimated under very local assumptions of the dynamics, through the estimation of the numerical derivative of $y$ and $u$". More details can be found in (Fliess and Join (2013)).

This question is linked to the previous one for which this citation has been added.

**Q2.8 :** **In Figure 6, the close-loop scheme is not clear. Why is the chopped wind flow signal is added with a signal coming out from "Blade section" block? The measured lift is produced by the blade section directly, but it is not produced by the wind flow directly. You should modify this diagram.**

We thank the reviewer for this remark. The diagram has been modified accordingly.

**Q2.9 : At line 186, what is "PWM"? This term is appeared in the first time, you need to spell the complete words.**

The definition of PWM has been added the first time it appears in the document as follows:

**"Pulse-Width Modulation"**

**Q2.10 : At line 210, "The PID control design requires at least to know a bit the dynamic of the system". What do you mean "a bit"? Can you specify what are the knowledge from the dynamic of the system is needed for the design of the PID control?**

Authors would like to thanks to reviewer for pointing out this misleading expression that has been rephrased as follows:

**"Regarding the robust PID control, it requires a transfer function modeling at some specifically operating points."**

**Q2.11 : At line 232, what is "SSE" and "STD"? These terms appear in the first time, you need to spell the whole words.**

Thank you for noticing, definitions were in the footnote p14. It has been introduced earlier, at the start of section 4.2 as follows:

**"the sum of square error (SSE), $SSE = \sum_k (e_k)^2$, the variation of the control input (VarU), $VarU = \sum_k |u_{k+1} - u_k|$ and the usual standard deviation of the output y, $STD=\sqrt{Variance(y)}$."**

**Q2.12 : At line 278, Scenario 2 is selected, but it has the lowest turbulent intensity. Is this a proper scenario which can demonstrate the performance of all the controller? Can you explain this?**

All scenarios have the same initial turbulence intensity as stated in section 2.1. The number indicated in percentage is the blockage ratio introduced by the chopper system. However, we agree with the reviewer that the external perturbations may impact the performances of the controllers. However, within the tested perturbations, we did not perceived significant differences with the average performances summarized in figure 7, and scenario #2 was arbitrarily chosen in the rest of the analysis to illustrate the operations. This is now stated as follows:

**"This histogram is found representative of all scenarios of table 1, even when different blockage ratios were set (scenario #2 and #3). Scenario #2 has been arbitrarily selected in the rest of the analysis to illustrate the averaged performance of figure 7."**

**Q2.13 : Figure 9, it seems that AST is not able to converge to the reference value. Can you explain this?**

We Agree with the reviewer that in Figure 9 (now figure 8), AST reaches the lift reference most of the time by extremes of the output signal but not always by its mean value. It may be attributed to a slight inexactitude in the quantification of the error in the AST real-time algorithm; however, it needs further investigation. This interesting remark has been added in the description of AST results as follows:

**"On can note that, contrary to other controllers, the lift reference is reached most of the time by extremes of the output signal and not always by its mean value. It may be attributed to a slight inexactitude in the quantification of the error in the AST real-time algorithm; however, it needs further investigation."**

**Q2.14 : Comparing Figure 10, 11, 12, the reviewer sees that the PID (c) controller shows only slightly better tracking performance on the lift. It makes more sense if you plot Figure 10, 11 and 12 on one figure so that the comparison can be made directly, and it also can save some pages. Please do this modification.**

We disagree with the reviewer, the PIDa gives a significantly higher SSE and VarU than the other PIDs that can be clearly seen by examining figure 10 and 11 and confirmed in figure 7. We did try to overlap these figures for fair comparison, the result was however unreadable. We thus believe separate plots are more suited. The detailed comparison asked by the reviewer is however available in figure 12 and 13 (now figure 13 and 14) for a limited time.

**Q2.15 : The author should present in more detailed the anti-windup algorithms used in both PID and AST in section 3.**

Thank you for this remark, a reference has been added where a detailed definition of the anti-windup algorithm can be found. It now reads:

**"The easiest Anti Windup algorithm is typically applied for the PID control \textcolor{blue}{(see e.g. \cite{Franklin}, chap. 9 of the sixth edition)}"**

**Q2.16 : Suggestion: The font size in Figures should be increased.**

Thank you for noticing, the font size has been increased in figures.

**Q2.17 : Again, Figure 16, 17, 18 should be plotted in one figure to achieve a better comparison.**

We thank the reviewer for this remark, however, as stated in question 2.14, we believe figures should stay apart for more clarity.

**Q2.18 : Could you indicate why there is no solution? If there is no solution, isn't it counter-proofing that the robust PID algorithm is actually not robust? Can you explain?**

We agree with the reviewer that this point deserves some clarification. The results coded in the R-Romuloc toolbox are conservative. If a solution is found, it is guaranteed to satisfy the robust performances for the given linear uncertain plant. If no solution is found it

is either because no such robust solution exists, or, that the conservative (pessimistic) method failed to find it. The more one asks in terms of robustness and the smaller is the set of possible controllers. As the size of the uncertainties grow (more very different models to be stabilized by the same unique controller), the size of the set of stabilizing gains shrinks. At some level, the algorithms fail to find a value in that set. It may be because the set becomes empty, but this cannot be assessed by the coded algorithms. To make the remark more adequate in the paper we changed it to:

**"Remark that, if one adds to the polytope (a) the model identified for 2.5 % for the inflow velocity of 21.3 m.s−1 the R-Romuloc toolbox fails to find a solution. This could be either because of the conservativeness of the coded method or because no such robust PID exists."**

**Q2.19 :** remarks added in the PDF by the reviewer have been taken into account.

**End of review #2**